

**Multifractal behaviour of the soil water content of a**
**vineyard in NW Spain during two growing seasons**
**J. M. Mirás-Avalos[1,2], E. Trigo-Córdoba[1], R. da Silva-Dias[3], I. Varela-Vila[3] and A.**
**García-Tomillo[3]**
[1]{Estación de Viticultura e Enoloxía de Galicia, EVEGA-INGACAL, Ponte San Clodio s/n,
32428, Leiro, Ourense, Spain}
[2]{Departamento de Riego, Centro de Edafología y Biología Aplicada del Segura, CEBAS-
CSIC, Campus Universitario de Espinardo, 31000, Murcia, Spain}
[3]{Área de Edafología y Química Agrícola, Facultad de Ciencias, Universidade da Coruña.
Campus A Zapateira s/n 15008 A Coruña, Spain}
Correspondence to: J. M. Mirás-Avalos (jose.manuel.miras.avalos@xunta.es)
**Abstract**
Soil processes are characterized by a great degree of heterogeneity, which may be assessed by
scaling properties. The aims of the current study were to describe the dynamics of soil water
content at three depths in a vineyard under rain-fed and irrigation conditions and to assess the
multifractality of these time data series. Frequency domain reflectometry (FDR) sensors were
used for automatically monitoring soil water content in a vineyard located in Leiro (Ourense,
NW Spain). Data were registered at 30-minute intervals at three depths (20, 40 and 60 cm)
between 14$^{th}$ June and 26$^{th}$ August 2011 and 2012. Two treatments were considered: rain-fed
and irrigation to 50% crop evapotranspiration. Soil water content data series obeyed power
laws and tended to behave as multifractals. Values for entropy ($D_1$) and correlation ($D_2$)
dimensions were lower in the series from the irrigation treatment. The Hölder exponent of
order zero ($\alpha_0$) was similar between treatments; however, the widths of the singularity
spectra, $f(\alpha)$, were greater under irrigation conditions. Multifractality indices slightly
decreased with depth. These results suggest that singularity and Rényi spectra were useful for
characterizing the time variability of soil water content, distinguishing patterns among series
registered under rain-fed and irrigation treatments.



## 1   Introduction

Soil water storage variability is strongly related with topographical, geological, edaphic and vegetation factors (Braud et al., 1995). These environmental factors and processes (rainfall, evapotranspiration, runoff) do not operate independently but as a conjunction of processes with nested and complex effects. Overall, this results in a distribution of soil water storage that varies as a function of the temporal and spatial scales. Therefore, similar to other soil properties and processes (Western and Blöschl, 1999; Zeleke and Si, 2006), soil water storage along time is a complex process characterized by a lack of homogeneity; heterogeneity in space and/or time is a feature that can be described by scaling procedures.

Fractals have been widely employed in soil science, as soil properties may be described through scale invariance concepts (Tyler and Wheatcraft, 1990; Perfect et al., 1996; Vidal Vázquez et al., 2007; Biswas et al., 2012a). More recently, several authors performed multifractal studies of heterogeneous time data series. For instance, Jiménez-Hornero et al. (2010) described ozone time series using the multifractal formalism. Rodríguez-Gómez et al. (2013) used a multifractal approach for characterizing solar radiation time series.

Soil water content can be automatically estimated by using sensors that measure variations in the soil dielectric constant, since it is strongly related with soil water content (Mestas-Valero et al., 2012). This parameter is characterized by its spiky dynamics, with sudden and intense peaks of high frequency activity, mostly at soil surface. Several studies have described scaling patterns for the behaviour of sol water content spatial distribution (e.g. Kim and Barros, 2002; Biswas et al., 2012b); however, multifractal analyses of continuously measured soil water content are scarce, except for a study on rain-fed grassland (Mestas-Valero et al., 2011). Therefore, the aim of the current work was to describe soil water dynamics in a vineyard subjected to two different treatments (rain-fed and irrigated) and to assess multifractality of these data series over two consecutive seasons.



**2 Materials and Methods**
**2.1 Description of the study area**
The experiment was conducted over two consecutive growing seasons (2011-2012) in a 0.2-
ha vineyard (*Vitis vinifera* L.) planted with cultivar 'Albariño', located in the experimental
farm of the Estación de Viticultura e Enoloxía de Galicia (EVEGA), in Leiro (42º 21.6' N, 8º
7.02' W, elevation 115 m), Ourense, Spain (Fig. 1). Vines were grafted in 1998 on 196-17C
rootstock and trained to a vertical trellis on a single cordon system (10-12 buds per vine).
Rows were east-west oriented, spacings between vines and between rows were 1.25 and 2.4
m, respectively (3333 vines ha$^{-1}$). The soil at the site was sandy-textured (64% sand, 16% silt,
20% clay), slightly acidic (pH 6.3), medium fertility (2.7% organic matter) and with a rather
shallow profile (≈1.2 m). The climate of the studied site is temperate, humid with cool nights
(Fraga et al., 2014).
**2.2 Experimental design**
The reference evapotranspiration ($ET_0$) per week for the site was calculated from weather
variables recorded at a station located 150 m away from the experimental vineyard using the
Penman-Monteith equation (Allen et al., 1998). The $ET_0$ was then used, along with a constant
crop coefficient ($K_c = 0.8$) to compute the amount of water required by the vines (Trigo-
Córdoba et al., 2015). Precipitation was substracted from $ET_c$ each week. The calculated
amount of water was applied the following week.
Treatments consisted of a rain-fed control and an irrigation to the 50% of $ET_c$. Irrigation was
applied from late June (after bloom) till mid-August, approximately two weeks prior to
harvest through two pressure-compensated emitters of 4 L h$^{-1}$ located 25 cm on either side of
the vine. Irrigation water was of good quality, with pH of 6.35, electrical conductivity of
163.4 µS cm$^{-1}$ and 0.4 mg of suspended solids. The water amount applied each season was 40
and 50 mm for 2011 and 2012, respectively (Table 1).
**2.3 Measurements**
The volumetric soil water content was continuously monitored through the soil profile in two
spots of the experimental vineyard (one in the rain-fed treatment and another in the irrigated
treatment) using two capacitance probes (EnviroSCAN, Sentek, Australia), based on the



frequency domain reflectometry (FDR) technique. Each probe was equipped with three
sensors installed on an access tube at 20, 40 and 60 cm depth and connected to a datalogger.
The probes were properly maintained for recording soil water content at half-hour intervals
over the 2011 and 2012 seasons. Here, data from the irrigation period (mid-June to late-
August) are reported.
In each treatment, the probe was located within two vines (Fig. 1), avoiding to be close to the
emitters (25 cm from the emitter and 50 cm from the vine trunk, approximately). The
equation provided by the manufacturer was used for transforming permitivity data registered
by the probes into soil water content.
## 2.4  Multifractal analysis
The concepts of multifractals and their estimation methods that were used in the current study
are next summarized. For detailed descriptions about multifractals, further information can be
found in Chhabra et al. (1989) and Everstz and Mandelbrot (1992).
To implement the multifractal analysis of one-dimensional soil water content time
distributions supported on a given interval $I$ = [a, b], a set of not-overlapping sub-intervals of $I$
with equal length is required. A common choice is to consider dyadic scaling down (Everstz
and Mandelbrot, 1992; Caniego et al., 2005), which means successive partitions of $I$ in $k$
stages ($k$ = 1, 2, 3…). Hence, at each scale, d, a number of segments, $N(\delta) = 2^k$ are obtained
with characteristic time resolution, $\delta = L \times 2^{-k}$, covering the whole extent of $I$.
Multifractal approach applied to time series has already been described (Jiménez-Hornero et
al., 2010), hence, we only summarize the technique used in the current study. The time
interval of soil water content data series, $L$, varied from half an hour to two months and the
minimum time resolution, $\delta_{ini}$, was chosen accounting for containing at least one half-hourly
averaged soil moisture data, $\theta_{ini}$, at every initial interval. According to this, the probability
mass distribution, $p_i(\delta)$, at time resolution $\delta$ was estimated as:
$$p_i(\delta) = \frac{\theta_i(\delta)}{\sum_{j}^{n_{ini}} (\theta_{ini})_j} \qquad (1)$$

where $\theta_i$ is the water content of the $i^{th}$ interval and $n_{ini}$ is the number of initial intervals with
mean soil water content $\theta_{ini}$.



The method of the moments was used (Chhabra et al., 1989) to analyze the multifractal
spectrum of the probability mass function, $p_i(\delta)$. The partition function $\chi(q, \delta)$ was estimated:
$$\chi(q,\delta) = \sum_{i=1}^{n} p_i(\delta)^q \qquad (2)$$
were moment $q$ is a real number between $-\infty$ and $+\infty$.
A log-log plot of the partition function versus $\delta$ for different values of $q$ yields:
$$\chi(q,\delta) \propto \delta^{-\tau(q)} \qquad (3)$$
were $\tau(q)$ is the mass scaling function of order $q$. The functions $f(\alpha)$ and $\alpha$ can be obtained by
Legendre transformation of the mass exponent, $\tau(q)$, as: $f(\alpha) = \alpha(q) - \tau(q)$ and $\alpha(q) = d\tau(q)/dq$,
respectively. Log-log plots of $\chi_q(\delta)$, versus $\delta$, however, typically exhibit linearity across a
limited scale range (e.g. Posadas et al., 2003), which results in drawbacks when using the
moment method to obtain the singularity spectrum.
The direct method (Chhabra and Jensen, 1989) avoid inaccuracies associated to the estimation
of $\alpha(q)$ by Legendre transformation. This method is based on the calculation of the
contributions of individual segments, $\mu_i(q,\delta)$, to the partition function, which are defined as:
$$\mu_i(q,\delta) = \mu_i^q(\delta) / \sum_{1}^{N(\delta)} \mu_i^q(\delta) \qquad (4)$$
Then, using a set of real numbers, $q$, $(-\infty < q < -\infty)$, the relationships applied to calculate $f(\alpha)$
and $\alpha$, can be expressed as:
$$f(\alpha(q)) \propto \frac{\sum_{i=1}^{N(\delta)} \mu_i(q,\delta) \log[\mu_i(q,\delta)]}{\log(\delta)} \qquad (5a)$$
and
$$\alpha(q) \propto \frac{\sum_{i=1}^{N(\delta)} \mu_i(q,\delta) \log[\mu_i(\delta)]}{\log(\delta)} \qquad (5b)$$
The $f(\alpha)$–$\alpha$ spectrum is reduced to a point for monofractal scaling type. The minimum scaling
exponent ($\alpha_{min}$) corresponds to the most concentrated region of the measure, and the
maximum exponent ($\alpha_{max}$) corresponds to the rarefied regions of the measure. A plot of $f(\alpha)$
vs. $\alpha$ is called multifractal spectrum. It is a downward function with a maximum at $q = 0$. The
width of the multifractal spectrum ($w = \alpha_{max} - \alpha_{min}$) indicates overall variability (Moreno et
al., 2008) similar to the nugget effects in geostatistics. For each data series, we calculated




multifractal spectrum with $q$ from –10 to +10 in steps of 0.5, fine enough to show the
multifractal behaviour in the studied moment range.
Multifractal measures can also be characterized on the basis of the generalized dimension, $D_q$,
of the moment of order $q$ of a distribution, defined by Grassberger and Procaccioa (1983),
based on the work of Rényi (1955). The $D_q$, of a multifractal measure is calculated as:

$$D_q = \frac{\tau(q)}{q-1} = \frac{1}{q-1} \lim_{\delta \to 0} \frac{\log|\chi_q(\delta)|}{\log \delta} \ , \ q \neq 1 \qquad (6a)$$

and

$$D_1 \approx \lim_{\delta \to 0} \frac{\sum_{i=1}^{n(\delta)} \mu_i(\delta) \log[\mu_i(\delta)]}{\log \delta}, \ q=1 \qquad (6b)$$

Equation (6a) shows that $\tau(q)$ is also related to the generalized fractal dimension, $D_q$. In fact,
the concept of generalized dimension, $D_q$, corresponds to the scaling exponent for the $q^{th}$
moment of the measure. Using equation (6a), $D_1$ becomes indeterminate. Therefore, for the
particular case that $q =1$, equation (6b) was employed.
For a monofractal, $D_q$ is a constant function of $q$. However, for multifractal measures, the
relationship between $D_q$ and $q$ is described by a S-shaped curve. In this case, the most
frequently used generalized dimensions are $D_0$ for $q = 0$, $D_1$ for $q = 1$ and $D_2$ for $q = 2$, which
are referred to as capacity, information (or Shannon entropy) and correlation dimension,
respectively. The information dimension, $D_1$, provides insight about the degree of
heterogeneity in the distribution of the measure. The correlation dimension, $D_2$, is associated
to the uniformity of the measure among intervals and describes the average distribution
density of the measure. In general, the generalized dimension, $D_q$, is more useful for the
comprehensive study of multifractals. Differences between $D_q$ allow comparison of the
complexity between measured soil water content data series. In homogeneous structures $D_q$
are close, whereas in a monofractal they are equal.



## 3 Results and discussion

### 3.1 Patterns of vineyard soil water content under rain-fed and irrigation conditions

Temperatures for the two studied growing seasons were similar in average (Table 1); however, rainfall and evapotranspiration were higher in 2012. Harvest date was almost the same in both years. Nevertheless, the temporal evolution of rainfall and $ET_c$ differed from year to year (Fig. 2), being greater during 2012, especially at the beginning of the study period. This fact caused a different scheduling of irrigation between years.

Soil water content decreased over the growing season under rain-fed conditions in both years (Fig. 3). However, when irrigation was initiated, soil water content became more stable in the irrigated treatment (Fig. 3). The magnitude of the soil water loss was more evident in the layers of 20 and 40 cm depth, and less important in the 60 cm layer, which may indicate the depth of the active root zone as well as the intensity of root water uptake at each soil layer, as reported for other cultivars and crops (Intrigliolo and Castel, 2009; Mestas-Valero et al., 2011), and proved that FDR probes can be successfully used for irrigation scheduling (Goldhamer et al., 1999), calibrating them with established indicators such as midday stem water potential (Mirás-Avalos et al., 2014). Indeed, our results suggest that the water amount applied through irrigation was enough for fulfilling vineyard water requirements over the two growing seasons studied.

### 3.2 Multifractality of the soil water content time series

Soil water content time series obeyed power law scaling, as shown by the double log plots (Fig. 4). These plots allow to identify the range of moments needed to describe the scale variation of the studied parameter (Vidal Vázquez et al., 2010).

Figure 4 shows the partition functions for rain-fed and irrigation conditions at 20 cm depth in 2011. Visually, a slight departure from the straight line model was observed for moments $q < -1$ (Fig. 4). In general, higher deviations from linearity were found for the highest $q$ moments in the data series from the irrigation treatment, when compared to those from the rain-fed treatment, especially in 2012. Nevertheless, determination coefficients, $R^2$, were greater than 0.9 for statistical moments in the range from $q = -10$ to $q = 10$, in all the studied data sets.



Consequently, scalings are adequately defined. Similar results were found by Mestas-Valero
et al. (2011) for soil water content under rain-fed grassland.
The $\tau(q)$ functions were different from a monofractal type of scaling for all series analyzed,
especially under irrigation conditions (Fig. 5), similar to results obtained by Biswas et al.
(2012b) for soil water storage. In fact, the hetereogeneity of the soil water content data series
from the irrigated treatment was greater than that of the rain-fed treatment (Fig. 5).
The value of $D_1$ is a good indicator of the heterogeneity degree in temporal distributions of a
given variable. The closer the $D_1$ value to $D_0$, the more homogeneous is the distribution of the
variable. In our case, rain-fed series were more homogeneous than the irrigated ones. In
general, soil water content recorded at 60 cm depth presented the lower differences between
$D_1$ and $D_0$ (Table 2), thus being more homogeneous both under rain-fed and irrigation
conditions. Moreover, 2012 data series presented a higher hetereogeneity than those from
2011 (Table 2) for both treatments, caused by the greater rainfall amount collected in 2012.
A monofractal would be characterized by $D_0 = D_1 = D_2$ (Evertsz and Mandelbrot, 1992). In all
the studied data series $D_0 > D_1 > D_2$ (Table 2), indicating that soil water content had a
tendency to behave as a multifractal. However, differences $(D_0 - D_1)$ ranged from 0.051 to
0.222 and $(D_1 - D_2)$ oscillated between 0.053 and 0.168, which suggests different degree in
the homogeneity/heterogeneity of soil water content depending on the treatment imposed and
the depth in the soil profile. In general, data series from the irrigation treatment showed
greater differences between $D_0$, $D_1$ and $D_2$ than the series from the rain-fed treatment for both
growing seasons. Moreover, the 60 cm depth layer presented smaller differences than the 20
and 40 cm layers (Table 2). The width of the $D_q$ spectra, determined by indicators such as $(D_0$
$- D_{10})$, showed different degrees of hetereogeneity, with a trend to decrease in depth and
under rain-fed conditions when compared with the irrigation treatment (Table 2). This is
caused by the spiky nature of soil water content and indicates a multiple scaling nature at
shallow depths.
Generalized dimensions, or Rényi spectra, calculated for the range between $q = -10$ and $q =$
10 for soil water content data series at three depths under rain-fed and irrigation conditions
are displayed on Fig. 6. All the data series studied showed Rényi spectra as asymmetric
sigma-shaped curves with more curvature for the negative values of $q$ than for positive ones
(Fig. 6). The left part of the curves is concave down and it changes to concave up on the right
of the vertical axis. In the case of the soil water content series from the rain-fed treatment, the



most curved spectra corresponded to the 40 cm depth data series, whereas for the irrigation
treatment, the most curved one was the 20 cm depth data series (Fig. 6). When compared
between treatments, Rényi spectra were more curved under irrigation conditions and the
estimation errors were also greater under this treatment (Fig. 6). These results confirmed the
higher hetereogeneity (multifractality) of the data series from the irrigation treatment when
compared to those from rain-fed.
Mestas-Valero et al. (2011) obtained monofractal distributions of soil water content time
series under grassland when measured at depths greater than 40 cm, in contrast with our
results. This disagreement is likely caused by the fact that grapevine root system reach greater
depths than that of grass and vines are capable of uptaking water from deeper soil layers.
Determination coefficients, $R^2$, were highest for moments $q = 0$ and $q = 1$ and diminished for
the other $|q|$ moments. In the case of $q = 10$, $R^2$ was greater than 0.97 and 0.95 in the rain-fed
and irrigated data sets, respectively. For $q = -10$, $R^2$ values for rain-fed and irrigated data
series were greater than 0.99 and 0.91, respectively (data not shown). Standard errors of $D_q$
values increased with increasing $|q|$ moments and they were much lower for right ($q > 0$) than
for left ($q < 0$) branch of the Rényi spectra (Fig. 6).
Parameter $\alpha_0$ from the singularity spectra ranged from 1.056 to 1.146 in the rain-fed treatment
and from 1.075 to 1.187 in the irrigated treatment (Table 3). The singularity spectrum allows
for analyzing similarity or difference between the scaling properties of the measures as well
as to assess the local scaling properties of soil water content measurements. The wider the
spectrum is (i.e., the largest $\alpha_{q-} - \alpha_{q+}$ value), the higher the heterogeneity in the scaling
indices and vice versa (Vidal Vázquez et al., 2010). Moreover, the $f(\alpha)$ spectrum branch
length gives insight about the abundance of the measure. Hence, small $f(\alpha)$ values at the end
of a long branch correspond to rare events.
Singularity spectra are characterized by a concave down shape (Fig. 7), showing an
asymmetrical curve with wider but shorter right side. Rain-fed data series showed a shorter
$f(\alpha)$ spectrum in both years, confirming their low degree of multifractality when compared to
the irrigated data series (Fig. 7).
Differences ($\alpha_{q-} - \alpha_0$ and $\alpha_0 - \alpha_{q+}$) indicate the deviation of the spectrum from its maximum
value ($q = 0$) towards the right side ($q < 0$) and the left side ($q > 0$), respectively (Vidal
Vázquez et al., 2010). Usually, soil water content data series from the rain-fed treatment





showed lower $\alpha_0 - \alpha_{q+}$ values than those from the irrigated treatment (Table 3). Moreover, the
highest values for this multifractal parameter were observed at 40 cm depth in both treatments
and years (Table 3). This may indicate that higher soil water contents were more frequent
under irrigation, being greater the differences between treatments at 40 cm depth in 2012. In
contrast, the right branch ($\alpha_{q-} - \alpha_0$) of the spectrum was usually wider for rain-fed conditions
(Table 3). These results confirm the differential homogeneity/heterogeneity pattern between
treatments evidenced by the generalized dimension, $D_q$, analysis (Table 2, Fig. 6).

## 4 Conclusions

Under the conditions of this study, continuous soil water content measurements at different
depths reliable described the soil water balance in a vineyard over two irrigation periods.
The logarithms of the partition function varied linearly with the logarithms of the time
resolution for all the studied depths under both treatments considered in the range of moments
$-10 < q < 10$, indicating that soil water content time series obeyed power laws.
The scaling properties of soil water content time series were reasonably fitted to multifractal
models. These properties were different for the rain-fed and irrigation treatments, implying a
higher heterogeneity for the data series from the irrigation treatment. Therefore, multifractal
analysis allowed us to discriminate among soil water content patterns in a vineyard for the
2011 and 2012 growing seasons as a function of irrigation use.

## Author contribution

J. M. Mirás-Avalos and E. Trigo-Córdoba designed and carried out the field experiment. J. M.
Mirás-Avalos, R. da Silva-Dias, I. Varela-Vila and A. García-Tomillo performed the
analyses. J. M. Mirás-Avalos prepared the manuscript with contributions from all co-authors.

## Acknowledgements

This work has been partialy supported by INIA (RTA2011-00041-C02-01), with 80% FEDER
funds. J.M. Mirás-Avalos thanks Xunta de Galicia for his 'Isidro Parga Pondal' contract. E.
Trigo-Córdoba thanks INIA for his FPI scholarship. The authors thank Dr. A Paz González
for support and discussion about multifractal analysis.



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



1  Table 1. Summary of climate variables (temperature, rainfall and $ET_0$), irrigation water

2  applied and harvest date for the studied period in 2011 and 2012 (from 14[th] June to 26

3  August).

| Year | Temperature (ºC) | | | Rainfall | $ET_0$ | Irrigation | Harvest date |
|------|---------|---------|---------|------|------|------|------|
|      | Minimum | Maximum | Average | (mm) | (mm) | (mm) |      |
| 2011 | 12.44 | 28.86 | 20.15 | 25.60 | 230.78 | 39.67 | 14[th] September |
| 2012 | 12.33 | 28.21 | 19.67 | 65.60 | 344.91 | 50.00 | 13[th] September |





1  Table 2. Selected multifractal parameters: generalized dimensions, for the first-three positive

2  moments, $D_0$, $D_1$, and $D_2$, with their respective errors of estimation, and two multifractality

3  indices $\Delta(D_0 - D_2)$ and $\Delta(D_0 - D_{10})$.

| Treatment | Depth (cm) | $D_0$ | $D_1$ | $D_2$ | $\Delta(D_0 - D_2)$ | $\Delta(D_0 - D_{10})$ |
|---|---|---|---|---|---|---|
| | | | 2011 | | | |
| Rain-fed | 20 | $0.999 \pm 0.001$ | $0.937 \pm 0.008$ | $0.884 \pm 0.016$ | 0.115 | 0.672 |
| | 40 | $1.000 \pm 0.000$ | $0.881 \pm 0.007$ | $0.746 \pm 0.014$ | 0.254 | 0.752 |
| | 60 | $1.000 \pm 0.000$ | $0.925 \pm 0.007$ | $0.868 \pm 0.013$ | 0.133 | 0.656 |
| | 20-60 | $1.000 \pm 0.000$ | $0.916 \pm 0.008$ | $0.833 \pm 0.019$ | 0.167 | 0.589 |
| Irrigated | 20 | $0.999 \pm 0.001$ | $0.868 \pm 0.013$ | $0.778 \pm 0.026$ | 0.221 | 0.757 |
| | 40 | $1.000 \pm 0.000$ | $0.852 \pm 0.019$ | $0.773 \pm 0.026$ | 0.227 | 0.698 |
| | 60 | $1.000 \pm 0.000$ | $0.852 \pm 0.022$ | $0.758 \pm 0.034$ | 0.242 | 0.664 |
| | 20-60 | $1.000 \pm 0.000$ | $0.861 \pm 0.023$ | $0.773 \pm 0.037$ | 0.227 | 0.695 |
| | | | 2012 | | | |
| Rain-fed | 20 | $0.999 \pm 0.001$ | $0.861 \pm 0.014$ | $0.771 \pm 0.025$ | 0.228 | 0.856 |
| | 40 | $1.000 \pm 0.000$ | $0.888 \pm 0.008$ | $0.739 \pm 0.017$ | 0.261 | 0.801 |
| | 60 | $1.000 \pm 0.000$ | $0.949 \pm 0.004$ | $0.907 \pm 0.005$ | 0.093 | 0.548 |
| | 20-60 | $1.000 \pm 0.000$ | $0.898 \pm 0.006$ | $0.768 \pm 0.016$ | 0.232 | 0.682 |
| Irrigated | 20 | $0.984 \pm 0.006$ | $0.831 \pm 0.010$ | $0.731 \pm 0.019$ | 0.253 | 1.024 |
| | 40 | $0.979 \pm 0.006$ | $0.757 \pm 0.014$ | $0.589 \pm 0.022$ | 0.390 | 1.210 |
| | 60 | $1.000 \pm 0.000$ | $0.907 \pm 0.007$ | $0.805 \pm 0.015$ | 0.195 | 0.622 |
| | 20-60 | $0.993 \pm 0.003$ | $0.822 \pm 0.016$ | $0.707 \pm 0.030$ | 0.286 | 1.085 |



Table 3. Selected multifractal parameters derived from the $f(\alpha)$ singularity spectra: most
positive ($q_+$) and most negative ($q_-$) limits the range of multifractal scaling, Hölder exponent
of order 0 ($\alpha_0$), most positive ($\alpha_{q+}$) and most negative ($\alpha_{q-}$) exponents, widths of the let ($\alpha_0 -$
$\alpha_{q+}$) and the right ($\alpha_{q-} - \alpha_0$) sides of the spectra.

| Treatment | Depth (cm) | $q_-$ | $q_+$ | $\alpha_0$ | $\alpha_{q+}$ | $\alpha_{q-}$ | $\alpha_0 - \alpha_{q+}$ | $\alpha_{q-} - \alpha_0$ |
|---|---|---|---|---|---|---|---|---|
| | | | | 2011 | | | | |
| Rain-fed | 20 | -1.5 | 3.5 | 1.066 | 0.768 | 1.339 | 0.299 | 0.273 |
| | 40 | -3.5 | 2 | 1.093 | 0.632 | 1.328 | 0.460 | 0.235 |
| | 60 | -3.5 | 2 | 1.087 | 0.718 | 1.403 | 0.369 | 0.315 |
| | 20-60 | -4 | 2 | 1.074 | 0.762 | 1.297 | 0.312 | 0.222 |
| Irrigated | 20 | -2.5 | 2 | 1.136 | 0.714 | 1.450 | 0.422 | 0.314 |
| | 40 | -4 | 3 | 1.160 | 0.664 | 1.383 | 0.496 | 0.222 |
| | 60 | -5 | 2 | 1.132 | 0.700 | 1.333 | 0.435 | 0.200 |
| | 20-60 | -4.5 | 2 | 1.142 | 0.709 | 1.375 | 0.433 | 0.233 |
| | | | | 2012 | | | | |
| Rain-fed | 20 | -2.5 | 3 | 1.146 | 0.659 | 1.526 | 0.487 | 0.380 |
| | 40 | -3.5 | 2 | 1.082 | 0.603 | 1.301 | 0.479 | 0.219 |
| | 60 | -2 | 5.5 | 1.056 | 0.746 | 1.296 | 0.309 | 0.240 |
| | 20-60 | -5 | 2 | 1.077 | 0.651 | 1.265 | 0.426 | 0.188 |
| Irrigated | 20 | -0.5 | 2.5 | 1.164 | 0.602 | 1.361 | 0.562 | 0.197 |
| | 40 | -1 | 1.5 | 1.187 | 0.575 | 1.491 | 0.611 | 0.304 |
| | 60 | -4 | 2 | 1.075 | 0.716 | 1.223 | 0.360 | 0.148 |
| | 20-60 | -1 | 2 | 1.172 | 0.624 | 1.489 | 0.548 | 0.317 |





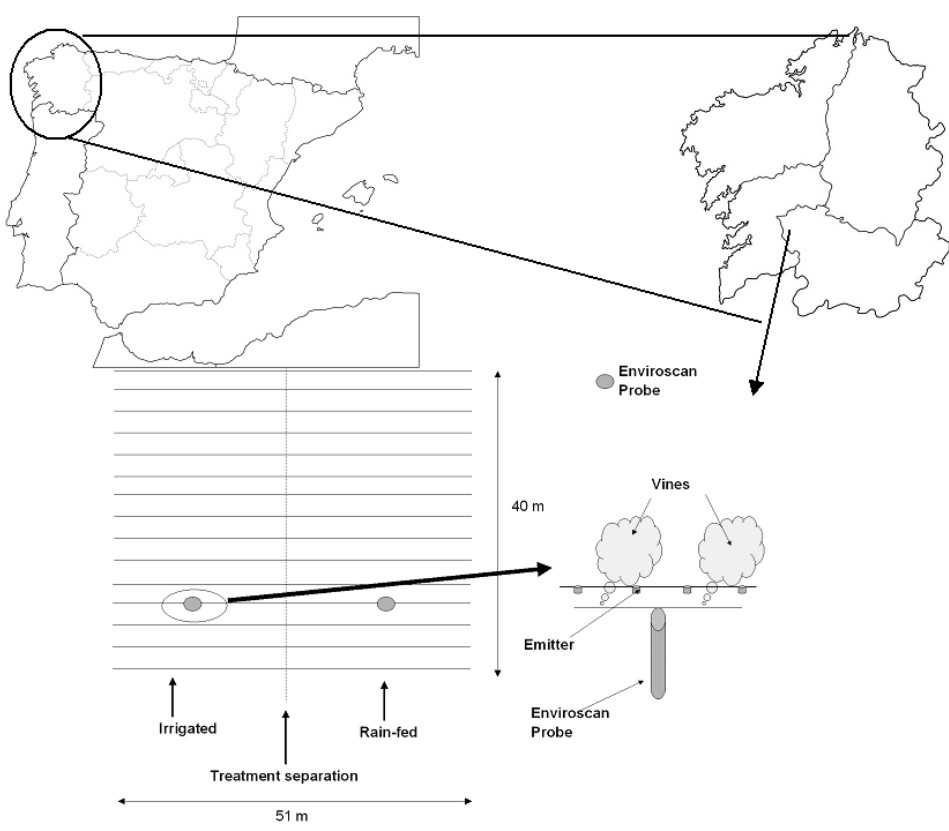

3    Figure 1. Location of the studied vineyard and experimental layout.



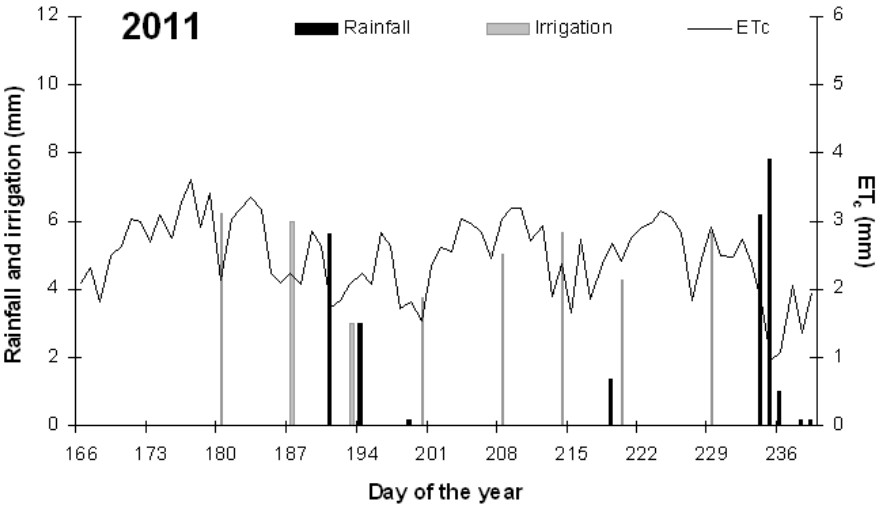

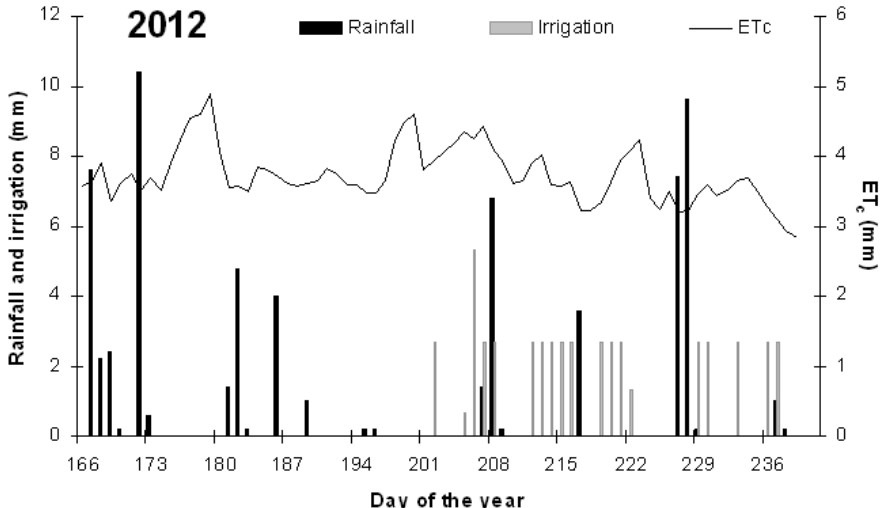

3    Figure 2. Crop evapotranspiration (ET$_c$), rainfall and irrigation water applied over the two

4    growing seasons studied, 2011 and 2012. Day of the year 166 is 14$^{th}$ June.





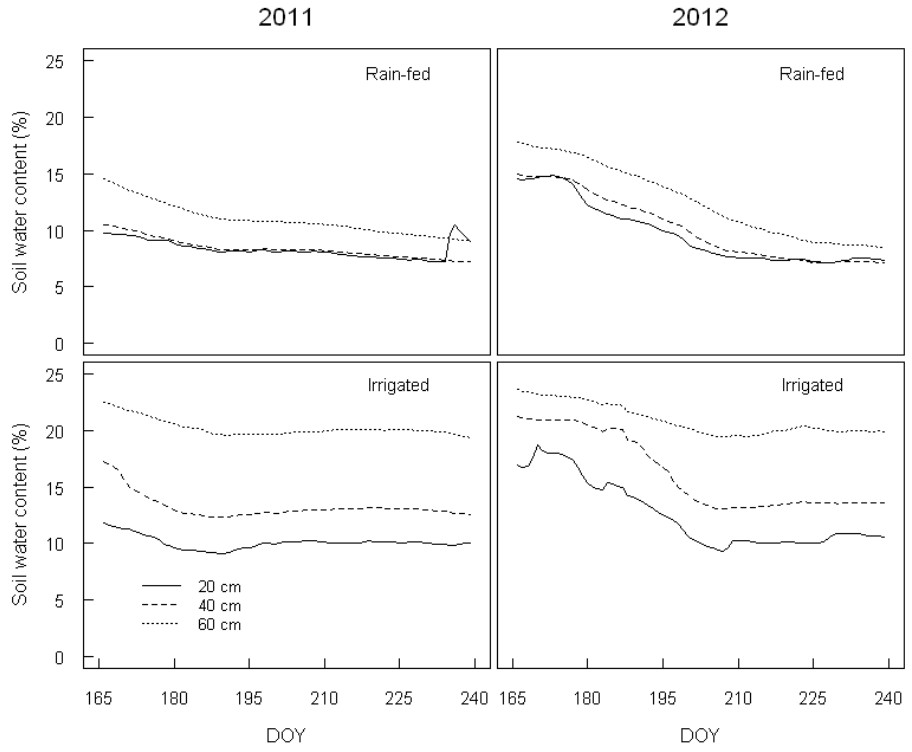

2    Figure 3. Soil water content at three depths (20, 40 and 60 cm) for rain-fed and irrigation

3    treatments over the 2011 and 2012 growing seasons. DOY stands for Day of the Year (165 =

4    13[th] June).





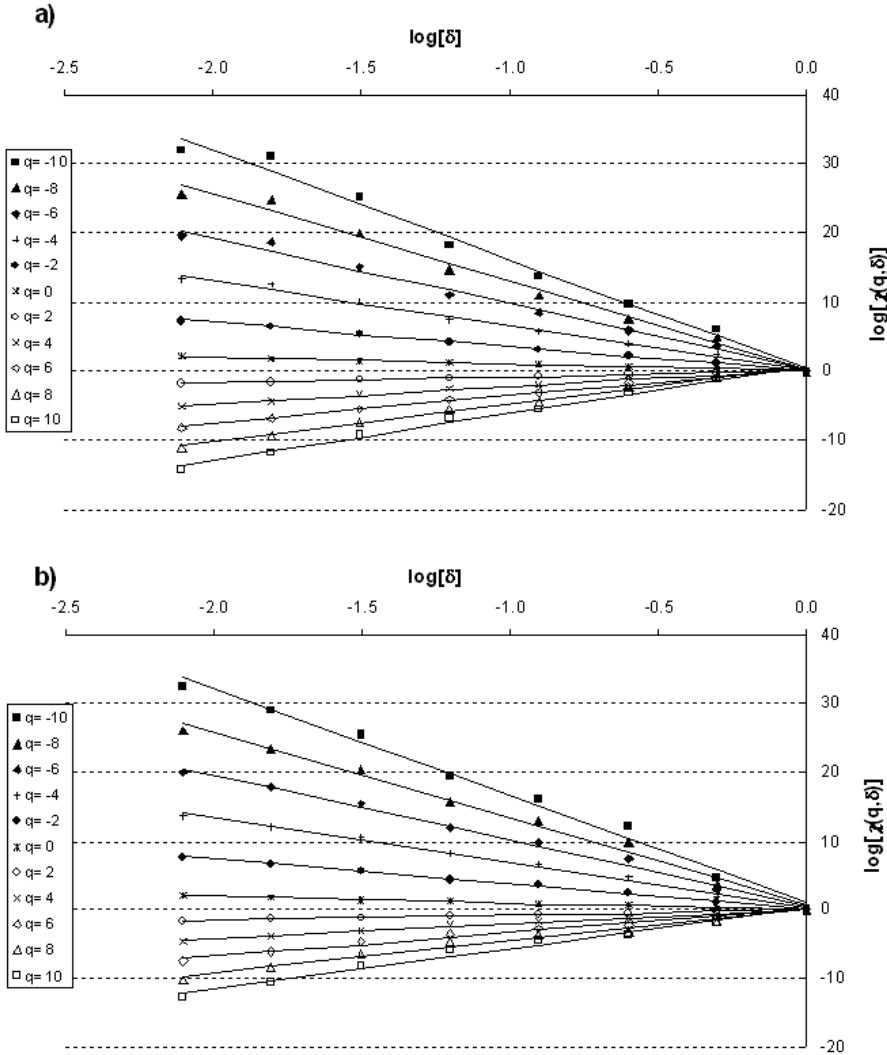

3    Figure 4. Selected plots of the natural logarithms of the partition function, $\chi(q,\delta)$, versus the

4    time resolution, $\delta$. a) rain-fed treatment at 20 cm depth in 2011; b) irrigated treatment at 20

5    cm depth in 2011.





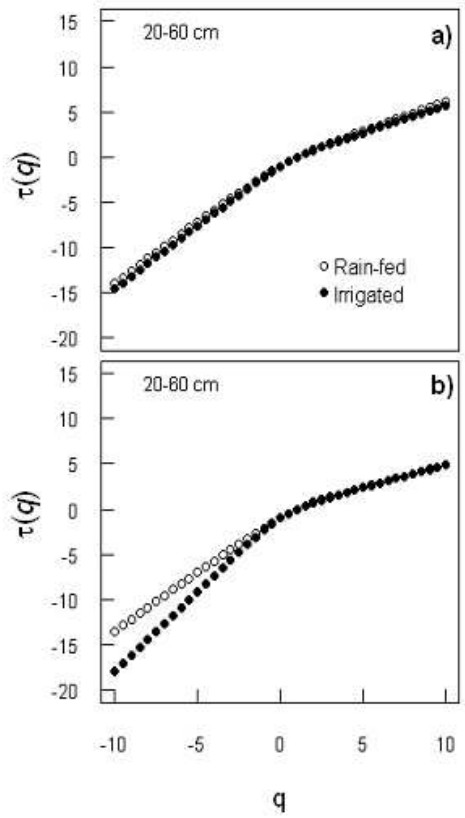

2    Figure 5. Mass exponents, $\tau(q)$, of soil water content averaged from 20 to 60 cm depth for

3    rain-fed and irrigation treatments: a) 2011, b) 2012.





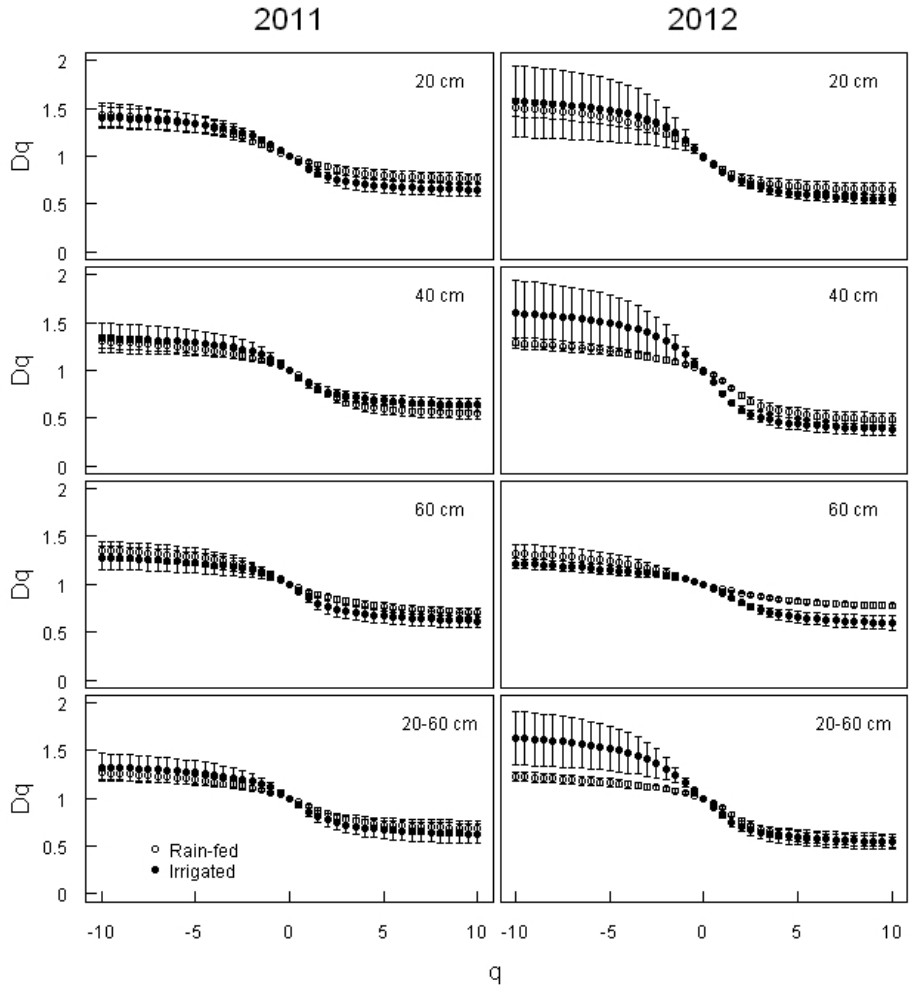

3     Figure 6. Generalized dimension, $D_q$, spectra (-10 < q < 10) of soil water content for rain-fed

4     and irrigation treatments at the studied depths in 2011 and 2012. Bars indicate estimation

5     errors.

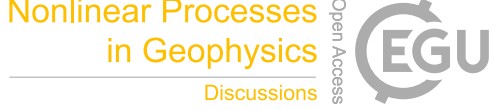



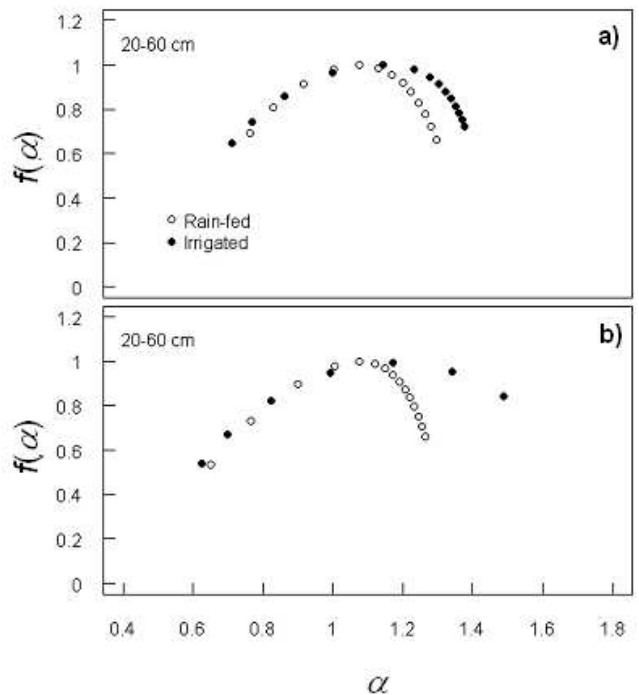

3    Figure 7. Singularity spectra for soil water content averaged from 20 to 60 cm depth for rain-

4    fed and irrigation treatments: a) 2011, b) 2012.