# Peer review of "J. M. Mirás-Avalos1,2, E. Trigo-Córdoba1, R. da Silva-Dias3, I. Varela-Vila3 and A. García-Tomillo3"

_Nonlinear Processes in Geophysics, 2016_

## Referee Comment (RC1) · M. G. Wilson (Referee) · 25 Apr 2016

M. G. Wilson (Referee)

mwilson@parana.inta.gov.ar

This paper is precise and concrete. The paper describes dynamics of soil water content and considers the multifractality of these time data series of a vineyard in NW Spain. The paper is showing scaling properties of soil water content time series, these properties were different for the rain-fed and irrigation use, and higher heterogeneity under irrigation conditions

---

## Referee Comment (RC2) · Anonymous Referee #2 · 13 Jun 2016

The aim of this paper is to describe soil water dynamics in vineyard and to assess multi-fractality . This topic is very interesting, because there is no application of multifractality to the temporal evolution of the soil water content in vineyard under two different water supply methods. This paper presents on an interesting subject and well written. The multifractality is analysed using the classical method of moments A question will be if the authors have detected the effects of seasonal trend on the multifractal analysis Specific and technical comments

Page 3 line 18 when calculating the soil water budget would have to take into account both the interception, and surface storage, and consider surface runoff Page Line 16 "0.4 mg of suspended soils", units must be mg/l or other concentration units. Page

4 line 8 "..the equation provided by the manufactured was used for transforming permittivity data registered by the probes into soil water content" justify this statement , because FDR calibration strongly depends on soil type. Page 7 line 15, add "and soil evaporation" Page 7. Line17 "...Indeed, our results suggest that the water amount applied through irrigation was enough for fulfilling vineyard water requirements over the two growing seasons studied" justify this statement Table 1 Use ETc data, I don't know if the depth of irrigation dose is net or gross. The dose of irrigation calculated seems very low, because if we calculate the approximate dose irrigation as precipitation less than 50% of crop evapotranspiration assuming negligible value interception, surface storage and surface runoff, the value obtained would be much higher than the value shown in table Fig 3 year 2011 around day 230 there is increase in rain-fed treatment in 20 cm depth but this increase is not showed in irrigated plot. Fig 5 and 7. Improve figure quality

---

## Author Comment (AC1) · 23 Jun 2016

Leiro, 23.06.2016

Dr. Antonio Paz González

Editor Special Issue "Multifractal Analysis in Soil Systems"

Nonlinear Processes in Geophysics

Subject: Submission of revised manuscript to *Nonlinear Processes in Geophysics*

Dear Editor,

On behalf of all authors, I would like to submit the following manuscript entitled: "Multifractal behaviour of the soil water content of a vineyard in NW Spain during two growing seasons", by José Manuel **Mirás-Avalos**, Emiliano **Trigo-Córdoba**, Rosane **da Silva-Dias**, Irene **Varela-Vila**, and Aitor **García-Tomillo**.

We thank all the comments and suggestions posed by the reviewers. All comments were taken into account and changes according to them were performed throughout the text. A specific answer to all the reviewers' comments is provided at the end of this letter. Besides, we moved a table and two figures to supplementary materials in order to alleviate manuscript length. We also corrected several mistakes in the reference list.

We hope that, at its present form, our manuscript would reach the high quality standards for publication on *Nonlinear Processes in Geophysics*.

Looking forward to hearing from you.

Sincerely yours,

*José Manuel Mirás Avalos*
Researcher

Estación de Viticultura e Enoloxía de Galicia (EVEGA)

Ponte San Clodio s/n, 32428, Leiro (Ourense, Spain)

Email: jose.manuel.miras.avalos@xunta.es

Reviewer: 1 (M.G. Wilson)

This paper is precise and concrete. The paper describes dynamics of soil water content and considers the multifractality of these time data series of a vineyard in NW Spain. The paper is showing scaling properties of soil water content time series, these properties were different for the rain-fed and irrigation use, and higher heterogeneity under irrigation conditions

*Thank you very much for your comment, Dr. Wilson.*

Reviewer: 2

The aim of this paper is to describe soil water dynamics in vineyard and to assess multifractality. This topic is very interesting, because there is no application of multifractality to the temporal evolution of the soil water content in vineyard under two different water supply methods. This paper presents on an interesting subject and well written. The multifractality is analysed using the classical method of moments A question will be if the authors have detected the effects of seasonal trend on the multifractal analysis

*Thank you very much for your comments. Regarding your question about the effects of seasonal trend on the multifractal analysis, we are not sure about what you mean. If you refer to differences in multifractal parameters from one season to the other; we addressed this on the former version by stating that "2012 data series presented a higher hetereogeneity than those from 2011" (page 8, lines 13-14). In the current version of the manuscript we added "Moreover, the width of the Dq spectra increased from 2011 to 2012 in both treatments, mainly in the 20 and 40 cm depths" (page 8, lines 27-28). We also added "Our results showed that the width of the singularity spectra increased in both treatments from 2011 to 2012 (Table 2)" (page 9, lines 26-27). Finally, in the conclusions section, we added ", which tended to increase in the second year of the study (2012)" (page 10, lines 21-22). We hope that these additions answer your question.*

Specific and technical comments

Page 3 line 18 when calculating the soil water budget would have to take into account both the interception, and surface storage, and consider surface runoff

*Yes, you are right, but we did not consider surface runoff because it was negligible on the studied plot. Surface storage and water interception by the canopy were not accounted for because our objective was not calculate a precise soil water budget but provide irrigation according to midday stem water potential readings and crop evapotranspiration, as described in this portion of the manuscript.*

Page Line 16 "0.4 mg of suspended soils", units must be mg/l or other concentration units.

*Yes, you are right, it is mg/L. We corrected it in the revised version of the manuscript.*

Page 4 line 8 "..the equation provided by the manufactured was used for transforming permittivity data registered by the probes into soil water content" justify this statement, because FDR calibration strongly depends on soil type.

*Yes, you are right, soil type greatly influences the FDR probe readings. However, we used the equation provided by the manufacturer because we did not use these measurements for scheduling irrigation since we relied on stem water potential readings. Besides, the default equation can be used for relative or differential measurements since we are comparing the performance of two probes in the same soil but under two different irrigation regimes. We included a brief text explaining this and providing a citation from a work by Paraskovas et al. (2012) in International Agrophysics in order to justify our procedure.*

Page 7 line 15, add "and soil evaporation"

*Added as suggested.*

Page 7. Line17 ". . .Indeed, our results suggest that the water amount applied through irrigation was enough for fulfilling vineyard water requirements over the two growing seasons studied" justify this statement Table 1 Use ETc data, I don't know if the depth of irrigation dose is net or gross. The dose of irrigation calculated seems very low, because if we calculate the approximate dose irrigation as precipitation less than 50% of crop evapotranspiration assuming negligible value interception, surface storage and surface runoff, the value obtained would be much higher than the value shown in table

*We removed the sentence because we did not provide data on grapevine vegetative growth, yield and berry quality, which were slightly affected by irrigation and made us to conclude that. These data are pending for publication and have been submitted to another journal; therefore, we considered removing this sentence as the best option since it does not affect the main focus and conclusions from the current manuscript.*

*We agree with you in the sense that Table 1 led to confusion since the data reported there referred to the whole period of measurements and not exclusively to the irrigation period. Besides, this table provided similar information to that displayed in figure 2. Therefore, we removed the table from the main manuscript and moved it to supplementary materials and added the information referred to the irrigation period for each year.*

Fig 3 year 2011 around day 230 there is increase in rain-fed treatment in 20 cm depth but this increase is not showed in irrigated plot.

*This is true; it was caused by three rainfall events happening on consecutive days from the day 234 to the day 236, accumulating 15 mm. The 20 cm depth sensor in the rain-fed treatment responded quickly to this increase (increases in soil water content were observed in day 235), whereas the*

*sensor in the irrigated treatment responded more slowly (day 237). Moreover, when checking the data, we observed that this increase in the soil water content at 20 cm depth in the rain-fed treatment lift the values up to those observed in the irrigated treatment at the same depth. Therefore, it seemed to be just a delay in the response of the sensor since the water content at 20 cm depth in the irrigated treatment also increased (in a lower proportion than that in the rain-fed treatment) after two days of the last rainfall event.*

Fig 5 and 7. Improve figure quality

*Thank you, quality of the figures has been improved as you suggest. However, in order to reduce manuscript length, we moved figure 5 to the supplementary material.*

[revised manuscript text omitted]

**Supplementary information**

**Table S1.** Summary of climate variables (temperature, rainfall, $ET_0$ and $ET_c$), irrigation water applied and harvest date for the studied and irrigation periods in 2011 and 2012.

| | | 2011 | 2012 |
|---|---|---|---|
| Measurement period | | 14 June to 26 August | 14 June to 26 August |
| Temperature (ºC) | Maximum | 28.86 | 28.21 |
| | Minimum | 12.44 | 12.33 |
| | Mean | 20.15 | 19.67 |
| Rainfall (mm) | | 25.60 | 65.60 |
| $ET_0$ | | 230.78 | 344.91 |
| $ET_c$ | | 184.63 | 275.93 |
| | | | |
| Irrigation period | | 9 July to 16 August | 20 July to 22 August |
| Temperature (ºC) | Maximum | 29.14 | 30.37 |
| | Minimum | 12.61 | 12.77 |
| | Mean | 20.36 | 20.82 |
| Rainfall (mm) | | 10.20 | 29.20 |
| $ET_0$ | | 117.86 | 162.52 |
| $ET_c$ | | 94.28 | 130.02 |
| Irrigation (mm) | | 39.67 | 50.00 |
| Harvest date | | 14[th] September | 13[th] September |

**Figure S1.** Selected plots of the natural logarithms of the partition function, $\chi(q,\delta)$, versus the time resolution, $\delta$: a) rain-fed treatment at 20 cm depth in 2011; b) irrigated treatment at 20 cm depth in 2011.

[Figure]

**Figure S2.** Mass exponents, τ(q), of soil water content averaged from 20 to 60 cm depth for rain-fed and irrigation treatments in 2011 and 2012.

[Figure]

---

## Author Comment (AC2) · 23 Jun 2016

Thank you very much for your comments. In the attached file, we supplied an answer to all the queries raised by the reviewers.

Kind regards

Please also note the supplement to this comment:
http://www.nonlin-processes-geophys-discuss.net/npg-2016-14/npg-2016-14-AC2-supplement.pdf